# Handgrip Strength in Young Adults: Association with Anthropometric Variables and Laterality

**DOI:** 10.3390/ijerph17124273

**Published:** 2020-06-15

**Authors:** Luciana Zaccagni, Stefania Toselli, Barbara Bramanti, Emanuela Gualdi-Russo, Jessica Mongillo, Natascia Rinaldo

**Affiliations:** 1Department of Biomedical and Specialty Surgical Sciences, Faculty of Medicine, Pharmacy and Prevention, University of Ferrara, 44121 Ferrara, Italy; luciana.zaccagni@unife.it (L.Z.); mngjsc@unife.it (J.M.); natascia.rinaldo@unife.it (N.R.); 2Biomedical Sport Studies Center, University of Ferrara, 44123 Ferrara, Italy; 3Department of Biomedical and Neuromotor Science, University of Bologna, 40126 Bologna, Italy; 4University Center for Studies on Gender Medicine, University of Ferrara, 44121 Ferrara, Italy

**Keywords:** handgrip strength, anthropometry, handedness, body composition, physical activity, sports practice

## Abstract

The measurement of handgrip strength (HGS) is an indicator of an individual’s overall strength and can serve as a predictor of morbidity and mortality. This study aims to investigate whether HGS is associated with handedness in young adults and if it is influenced by anthropometric characteristics, body composition, and sport-related parameters. We conducted a cross-sectional study on a sample of 544 young Italian adults aged 18–30 years. We measured HGS using a dynamometer and collected data on handedness and physical activity, along with anthropometric measurements. In both sexes, the HGS of the dominant side was significantly greater than that of the non-dominant side. Furthermore, in ambidextrous individuals, the right hand was stronger than the left. A comparison between the lowest and the highest tercile of HGS highlighted its significant association with anthropometric and body composition parameters in both sexes. Moreover, sex, dominant upper arm muscle area, arm fat index, fat mass, and fat-free mass were found to be significant predictors of HGS by multiple regression analysis. Our findings suggest that HGS is especially influenced by body composition parameters and handedness category. Therefore, HGS can be used as a proxy for unhealthy conditions with impairment of muscle mass, provided that the dominance in the laterality of the subject under examination is taken into account.

## 1. Introduction

Handgrip strength (HGS) is a fundamental parameter in biomechanical modeling, which has found many important applications in the development of ergonomic tools, in the design of equipment and consumer products, and in sports practices [1,2,3,4,5,6,7,8]. Grip strength is crucial for the human body when performing prehensile and precision hand functions [9], and it is used as one of the main indicators for testing muscle power. Moreover, it is a low-cost tool for predicting an individual’s overall strength, which can reflect general health conditions and level of physical activity [10]. In fact, muscle weakness and low grip strength have been related to disability and are considered predictors of sarcopenia, higher recovery time, and higher mortality, especially in hospitalized patients [11,12,13,14]. Nevertheless, several populations lack the HGS reference values that are necessary to ensure the health, safety, comfort, and productivity of workers and consumers, as well as for clinical purposes and to monitor recovery after post-injury treatment [15].

Reference values are essential for the correct interpretation of acquired data, for the definition of appropriate treatments, and for the evaluation of the effectiveness of interventions [16,17,18]. Furthermore, reference values can also be used as motivation for patients during rehabilitation interventions. One of the goals of this study is to contribute new mean HGS values obtained from healthy young Italian individuals disaggregated by sex, which can be used as a reference for equivalent populations.

Many studies have reported the relationships between grip strength and various variables [9,19,20]; however, the predominant factor which influences hand grip strength remains unclear. From a summary of the principal findings in the literature, some relations can be identified. Age has an important influence on grip strength, for which a curvilinear relationship has been observed, resulting in an increase in grip strength with age that reaches a peak between 30 and 40 years of age, and decreases thereafter. The non-linear relation differs between the sexes and is more pronounced in females. Males maintain their grip strength for at least another decade longer than females [15,21,22,23]. The difference between the sexes is evident at all ages and males generally have greater grip strength than females [15,17,23,24,25]. This is the consequence of the larger size and related muscle mass in males, as muscle strength is a function of these characteristics. Males generally feature bigger arm muscles and are more involved in more activities which require strength than females. Strong correlations have also been reported between grip strength and body dimensions (e.g., weight, height, and hand length) [26,27,28,29,30,31,32]. With respect to body mass index, the results are controversial: some authors have reported that static grip strength is positively related to body mass index (BMI), considering it to be a predictor for grip strength, while others found no significant association, concluding that BMI does not influence handgrip strength [15,20]. Forearm circumference, wrist joint circumference, palm circumference, hand length, and middle finger length play major roles in influencing the dominant hand’s grip strength [15,20,33,34,35]. Working activity can also influence handgrip strength, where manual workers have a higher average grip strength than non-manual workers [15]. In light of the above, it is necessary to find the best predictors of HGS values among different anthropometric characteristics, as well as establish the impact of physical and sporting activities on hand strength.

Hand grip differs according to the laterality, where the dominant hand has a greater capability for grip strength than the non-dominant hand [15]. This is probably related to the fact that the dominant hand is used forcefully more often than the non-dominant one, such that the muscles of the dominant side are bigger and, thus, stronger. Nevertheless, this is true for people with a dominant right hand; while for people with a dominant left hand, this difference is smaller and sometimes not even significant [15,20,36]. In this respect, it is important to remember that humans demonstrate functional differences in the right and left of each bilaterally symmetrical body part [37,38,39,40,41]. Laterality is particularly evident in the functions of the fingers, such as for spoon use or letter writing, and has been considered to be due to the more preferential and frequent use of one hand in daily activities. Until now, lateral dominance of muscle function has frequently been reported. Laterality represents a multidimensional trait [42] and it is well-known that, in the adult population, 90% of people prefer to use their right hand for common manual tasks, whereas about 10% of the population is left-handed [43,44,45,46]. Another important observation is that, throughout human life, the development of laterality is a very active process, influenced by both genetic and environmental factors [46,47,48,49,50,51].

Handedness can be defined as “the individual’s preference to use one hand predominantly for unimanual tasks and the ability to perform these tasks more efficiently with one hand” [52]. Significant differences in the consistency of handedness between males and females have been observed, and a significantly higher number of ambidextrous individuals among males has been detected [51,53,54,55].

Another significant aim of this study is to assess the influence of laterality on hand muscle strength in young adults subdivided by sex, in order to evaluate whether the level of laterality assessed as the difference in handgrip skillfulness between the preferred and the non-preferred limb differs in males and females. Finally, considering the limited information on dominant handgrip strength in the literature [56], especially for youth, this study is designed to identify anthropometric and activity-related determinants of dominant hand strength in healthy young subjects.

## 2. Materials and Methods

### 2.1. Participants and Procedure

We carried out a cross-sectional survey which included young subjects, aged 18–30, in a convenience sample. Given the ethnic variability of the anthropometric traits [57,58], only individuals of Italian origin were included in the sample. They were recruited among the students of the Universities of Ferrara and Bologna (Northern Italy), according to the inclusion and exclusion criteria specified below.

The inclusion criteria for participation were: (1) being of Italian origin; (2) being aged ≥18 years; (3) having signed the written informed consent; and (4) having declared to be healthy. Those aged >30 years were excluded.

For this study, we excluded 16 foreign students, 20 students over 30 years of age, and 50 subjects with incomplete data on handedness. The result was a final sample of 544 young Italian adults (356 males and 188 females) with an average age of 21.6 ± 2.9 years for males and 21.3 ± 2.0 years for females.

The study was approved by the Ethics Committee of the University of Bologna (registration number: 253035, 17 October 2019) and followed the rules and principles of the Helsinki Declaration.

### 2.2. Data Collection

The collection of data was carried out through self-administered questionnaires on hand laterality and physical activity, as well as through the direct anthropometric measurements described in detail below.

Questionnaires: First, we collected personal demographic information (sex, place and date of birth, citizenship), health status (absence/presence of illness or infirmity), and the physical/sporting activity of the participants. With regards to sporting activity, participants were asked whether they had practiced any sport, which sport, for how many years, and how many hours they practiced per week. Sports were categorized according to their metabolic equivalent of task (MET) intensity values [59]. Physical activities were assessed by the International Physical Activity Questionnaire (IPAQ) and classified as light, moderate, or intense. We used the Edinburgh Handedness Inventory (EHI) as a method to assess handedness on a quantitative scale [60]. Participants examined the 10-item inventory by choosing the answer column (“left hand” or “right hand”) for each item and indicating their weak (1) or strong (2) preference for the left or right hand in performing the given action specified in the item. If there was no specific preference for one hand over the other in that item, the participant indicated 1 in both columns. At the end of the procedure, a handedness score (R) between −100 (strongly left) and +100 (strongly right) was obtained. An ambidextrous condition was indicated by scores between −40 and +40 [61,62].

Measurements: Anthropometric measurements were collected by trained anthropometrists using specific instruments (i.e., Raven anthropometer, Seca weighing scale, Lange plicometer, GPM measuring tape, and Takei handgrip dynamometer). This study took into consideration seven measurements—stature, weight, mid-upper arm circumference (MUAC), waist circumference (WC), triceps skinfold thickness, and right and left handgrip strength (HGS)—collected from each subject according to standard procedures [63,64,65], as well as five anthropometric indices (reported below) obtained from previous measurements. Moreover, the body density (BD), estimated according to the formula developed by Durnin and Womersley [66], was converted to percentage fat mass (%F) using the Siri [67] equation: ((4.95/BD) − 4.50) × 100. The fat mass (FM, kg) was calculated as (%F × weight)/100, and the fat-free mass (FFM, kg) as weight – FM.

With regards to anthropometric measurements, stature was measured from each subject while standing barefoot. The subject’s head was oriented according to the Frankfurt plane. The weight of each participant was measured, while dressed in underwear or light clothing, using a mechanical scale. The maximal HGS of both hands was measured in kg to five decimal points by a hand-grip dynamometer. The participants carried out three maximum voluntary contractions per side with a 60 s rest break between each test. In the statistical analyses, the highest strength value obtained in the three tests of each hand was used. MUAC and triceps skinfold thicknesses were measured on both sides of the body at the midpoint between the acromion process of the scapula and the olecranon process of the ulna. WC was measured at the narrowest point between the lowest ribs and the iliac crest.

For the anthropometric indices, the body mass index (BMI) was computed as weight (kg)/stature (m^2^). Subjects were classified as underweight, normal weight, overweight, or obese, according to the World Health Organization (WHO) cut-points [68]. Based on the %F value, subjects were classified as under-fat, normal fat, over-fat, or very over-fat, according to the cut-points of Gallagher et al. [69]. Using MUAC and triceps skinfold, we also calculated the following indices of nutritional status: total upper arm area (TUA, cm^2^), upper arm muscle area (UMA, cm^2^), upper arm fat area (UFA, cm^2^), and arm fat index (AFI, %) [70]. For all bilateral characteristics and related indices, we reported separately the mean values of the right, left, and dominant side (the last value was replaced by the mean of the two sides for ambidextrous subjects).

### 2.3. Statistical Analysis

Quantitative variables were described using means and standard deviations, and qualitative variables described by frequencies. Due to the fact that all the variables (except for skinfold thickness) had a normal univariate distribution (skewness and kurtosis were within acceptable ranges), parametric statistical tests were applied. Comparisons between two independent samples (males vs. females; lowest vs. highest tercile) were performed by Student’s *t*-test. These comparisons were carried out using the logarithmic transformation of skinfold thickness to normalize this trait, which is usual practice [66,71,72].

The paired samples (dominant vs. non-dominant handgrip strength) were compared with the *t*-test for dependent samples, or with the Wilcoxon signed-rank test when the subsample size for a certain trait was too small (i.e., comparison of HGS within some category of handedness). Then, one-way analysis of variance (ANOVA) was applied to detect any difference in HGS means of the dominant side among the three handedness groups (i.e., right-handed, left-handed, and ambidextrous individuals).

Comparisons among categorical data (frequencies of fat status categories; weight status categories) were performed with the Chi-squared test.

The Pearson correlation coefficient was used to measure associations of anthropometric traits with grip strength. Backward multiple regression analysis was carried out to assess possible predictors of dominant HGS. Anthropometric variables (of the dominant side for characters detectable on both sides), metabolic equivalent of task (MET), and fat status were included in the regression models in the continuous scale, while sex was included in the model as a binary variable with females as the reference group. Predictors input into the model were those found to have significant associations with HGS (i.e., *p* < 0.05), while those with *p* > 0.05 were removed from the model. The multicollinearity of the data was assessed by variance inflation factors (VIFs), assuming VIF values between 0.10 and 10 as acceptable [23].

A value of *p* < 0.05 was accepted as the level of significance for all statistical tests.

All analyses were performed using the software STATISTICA, version 11 (StatSoft, Tulsa, OK, USA).

## 3. Results

Table 1 shows descriptive statistics of the sample by sex. As expected, males were significantly taller, heavier, stronger in both hands in absolute and relative terms (handgrip strength/weight), and with larger circumferences (waist and upper arms), on average, but with less skinfold thickness than females.

Mean BMI values fell into the normal weight status, according to WHO weight status categories, but the sexes differed in their distribution of weight status: underweight and normal weight categories were more represented in females, while overweight and obese were more represented in males.

Concerning body composition, males were on average leaner than females: the muscle area of the upper arm and FFM in males were higher than in females, but the fat area of the upper arm, arm fat index, %F, and FM in males were lower than in females.

The fat status distribution also differed significantly by sex: the under-fat category was more represented in females (almost twice as often in females than in males), and over-fat and very over-fat categories were more represented in males (over-fat: almost twice as often in males as in females; very over-fat: no females).

In both sexes, the number of years of sports practice (about nine) were similar, but males practiced a significantly higher weekly amount of sport than females. Both sexes chose a similar sports discipline, according to the classification of METs: nearly three-quarters chose intense sports and almost one-fifth of them chose moderate sports.

With regards to physical activity (PA), as assessed with the IPAQ, males were significantly more active than females and practiced more intense activities than females.

Table 2 shows the results obtained from the administration of the EHI. Females had, on average, a significantly higher R score than males. The distribution of handedness differed significantly by sex, with more ambidextrousness among males and more right-handers among females.

Table 3 shows the results of the HGS by handedness category. In both sexes, the HGS of the dominant side was significantly greater than that of the non-dominant side, apart from left-handed females; moreover, in ambidextrous subjects, the right hand was stronger than the left hand.

No difference was detected in the comparison of the mean HGS of the dominant side among the three independent handedness groups (disaggregated by sex) by ANOVA (males: F = 1.010, df = 2; 358, *p* = 0.3653; females: F = 1.330, df = 2; 191, *p* = 0.2669).

Table 4 and Table 5 report the results of the comparisons between the lowest and the highest terciles of the dominant HGS, separately for each sex.

As shown in Table 4, males with higher values of HGS (highest tercile) were taller and heavier, with higher mean BMI and waist circumference values. Analyzing the body composition of the participants in detail, we found higher values of strength in males with higher values of upper arm muscularity, MUAC, FM, and FFM. However, the triceps skinfold, %F, and AFI did not differ between the lowest and the highest terciles. Consistently with the mean values of BMI, the distribution of weight status categories was significantly different between the two terciles, as we found high percentages of overweight and obese males in the highest tercile. In contrast, the fat status category distribution was similar between the two groups. Regarding sport and physical activity, only the years of sports practice and the intensity of the sport resulted in a significant difference between the two terciles. Males from the lowest tercile, who showed more years of experience and practiced sports classified as more intense by METs, had lower values of handgrip strength. There were no significant differences in the handedness distribution between the two terciles (Table 4).

Comparing the anthropometric and body composition characteristics between the lowest and the highest tercile in females (Table 5), the results were similar to those of the male group. Females with higher values of HGS were taller and heavier, with higher values of BMI, WC, triceps skinfold, MUAC, FM, FFM, UFA, and UMA. As in the male group, %F and AFI did not differ between the two terciles. Moreover, the sample distribution in the weight and fat status categories was similar between the two terciles. Considering the practice of sports and PA, only the hours of sports practice resulted in a significant difference between the two terciles, with females with higher values of HGS practicing more h/week. The handedness distribution was similar between the two terciles.

Table 6 shows the results of the correlation between the anthropometric variables and the dominant HGS for each sex. Almost all the traits were positively correlated with HGS in both sexes. HGS was positively correlated with dominant UFA in females and negatively correlated with AFI in males. Both FM and FFM proved to be positively correlated with HGS in both sexes.

Table 7 shows the multiple linear regression analysis for the whole sample, conducted with all the significant predictors input into the model with a backward method. In the multivariate model, sex, dominant UMA, AFI, FM (negative coefficient), and FFM were significant predictors of HGS. These independent variables accounted for 74.6% of the variance of HGS in young adults.

## 4. Discussion

The main purpose of this study was to analyze the relationship of HGS with anthropometric characteristics and laterality, taking physical activity and sports practice of the participants into account, in order to identify the determinants of dominant HGS by multivariate analysis. Moreover, we intended to analyze the differences in the values of dominant and non-dominant handgrip strength among three categories of handedness, comparing the results between the two sexes.

While the curvilinear association between HGS and age is well-known [73,74], the relationships between HGS and several anthropometric characteristics or indices have not been well documented in samples, nor selected for a specific sport, which has led to inconsistent or incomplete results in different studies. The association between BMI and strength, for example, has been widely debated in the literature; while some studies have recognized their positive association in both sexes, others have denied it [75,76,77]. In addition, previous analyses have not generally taken into account manual dominance, assessed according to defined criteria, as indicated by Oldfield [60]. Therefore, we conducted the present study on a sample of healthy young Italian adults of both sexes to examine these aspects in depth, taking into account their handedness through the EHI questionnaire. In particular, we observed significant differences in HGS between sides in the sample’s paired comparison after its division into three handedness categories by sex. The vast majority of the sample was right-handed, but there was a significantly different distribution of handedness categories between sexes: more males were left-handed (+0.8%) or ambidextrous (+12.5%), while more females were right-handed (+13.3%). This finding was consistent with previous studies [78,79].

Within the categories with different handedness, the results of this study indicate a significant difference in HGS between the sexes in favor of males, as well as a trend for the dominant hand to be significantly stronger in both sexes (this difference was not significant in left-handed females, probably due to the small sample size). Ambidextrous subjects of both sexes showed higher HGS of the right hand (significant only in the male sex). This latter trend is likely due to the greater pressure of our society to use the right hand in various occupational tasks from childhood on [47,80,81].

The HGS mean values of the dominant hand obtained in our study for young healthy Italian adults were 45.7 ± 8.2 kg for males and 28.9 ± 4.7 kg for females. A detailed comparison of our HGS results with previous studies is difficult, especially due to the different methods applied (if any) in the evaluation of handedness. In general, we observed that male and female HGS values were, on average, within the range of healthy young adults (not selected for sports) found in the literature (i.e., in subjects aged 20, in Innes [82]; 18–33, in Nicolay and Walker [83]; 20–24, in Werle et al. [84]; 18–25, in Koley [85]; 20, in McGrath et al. [86], 50th percentile values) with mean values ranging from 20.4 to 35.6 kg for the dominant hand among females, and from 39.5 to 58.8 kg for the dominant hand among males. A majority of studies reporting reference values for HGS were conducted by examining participants living in developed countries, mainly Europe, Australia, and the US [82,84,86]. Only a few studies have reported reference HGS values for developing countries (i.e., Werle et al. [84] for the Indian population). According to Dopsaj et al. [87], differences in HGS between sexes may be explained by various factors, such as the cross-sectional area, muscle fiber characteristics, amount of skeletal muscle mass, distribution of muscle mass in the upper limbs, and common anatomical differences.

According to the scientific literature [88], there is a strong positive correlation between HGS and weight. As body mass can be considered a confounding factor in the evaluation and analysis of HGS, we decided to adjust HGS to the body mass, in order to reduce this bias. Despite the fact that the importance of the adjustment of HGS for weight has been proven, especially for the association of this parameter with metabolic syndrome and sarcopenia [89], the results of our study did not show particular differences between the sexes using the two parameters of HGS; males had a stronger handgrip strength than females, regardless of their weight.

The dominant hand is used more often than the non-dominant one, such that the muscles of the dominant arm are bigger and stronger. The general rule assumes that the dominant hand is 5–10% stronger, compared to the non-dominant [90,91,92,93]. According to Adam et al. [93], there is a higher percentage of recruitment of motor units at lower absolute force levels in the dominant hand; whereas, in the non-dominant hand, there is a spread-out recruitment pattern. In our study, the difference was observed for the lowest percentages, shifting from 5.1% for left-handed females (5.9% in males) to a maximum of 5.8% for right-handed females (5.5% in males). These results are in line with the study of Ekşioğlu [15], and the possible differences may be due to the different methodologies used. In some studies, in fact, the dominance of the hand was established simply on the basis of the best performance obtained in the measurement of HGS and not on the basis of the function of the hands in different daily actions. In the examined ambidextrous subjects, the right hand was, on average, stronger than the left hand, with a difference of 2.9% in females and 4.5% in males. To explain this feature, it is important to take into account that most everyday tools and appliances are designed for right-handed individuals. A left-hander who does not have access to left-handed instruments often has to adapt by making his/her right hand the dominant hand [94]. As a result, the right hand is exercised more often than the left, on a daily basis [91]. In addition, Ozcan et al. [95] reported that the laterality of manual function is less pronounced in left-handed people than in right-handed people.

The results of the comparison between the lowest and the highest tercile of the dominant handgrip strength for each anthropometric variable showed a significant difference for almost all considered traits. Individuals of both sexes with higher values of HGS (highest tercile) were taller and heavier, with higher mean BMI, WC, D triceps skinfold (only in females), dominant (D) MUAC, D TUA, D UMA, D UFA (only in females), FM, and FFM. The larger sizes of the strongest individuals were confirmed by the significant correlation of anthropometric characteristics with HGS. As the HGS increased, height, weight, BMI, WC, D MUAC, D triceps (in females only), D TUA, D UMA, D UFA (in females only), FM, and FFM increased. In males, D AFI decreased as HGS increased. Previous studies have generally considered and confirmed only the associations of HGS with stature, weight, and BMI [96,97,98].

The hypothesis of a positive association between HGS and lean body mass, which has been formulated in other studies only on the basis of an increase in BMI (especially in males) [99,100], was confirmed in this study, through the assessment of body composition parameters and indices.

As evidence that BMI is an unreliable measure of body fat and does not differentiate between weight changes due to an increase or decrease in muscularity or adiposity [101], we found higher percentages of overweight and obese males in the highest tercile of HGS than in the lowest one (36.4% vs. 13.6%). As further confirmation, neither D triceps skinfold, D UFA, D AFI, nor %F differed between terciles and were not significantly correlated with HGS. As a consequence, underweight and normal weight males were more frequent in the lowest HGS tercile than in the highest one (86.4% vs. 63.6%). This trend was consistent with the results in the literature, reporting that grip strength is lower in underweight individuals and increases across normal and overweight categories, but plateaus through obese BMI ranges (although obese individuals are still stronger than normal weight individuals), suggesting that strength declines only in extremely obese individuals [41,86]. This was also consistent with the positive association we found between HGS and FM. In females, the trend was similar but more shaded. Therefore, underweight and normal weight females were more frequent in the lowest HGS tercile (94.8% vs. 86.6%) and overweight and obese females in the highest HGS tercile (13.4% vs. 5.2%), without any statistical significance. This finding could be related to the lower representation of overweight and obese categories in females. Again, however, D AFI and %F did not differ between terciles and showed no significant correlation with HGS.

The comparison between the parameters of physical activity and sport practice between the two terciles showed different results in the two sexes. According to the literature, intensity and years of sports practice are both important for increasing HGS [102,103]. Surprisingly, we observed significantly fewer years of sports practice in the stronger males, in addition to a lower weekly amount of sports activity and a lower number of METs, although not significant. A possible explanation could be that the HGS of the participants in this study was associated more with the specificity of each sport, rather than with the intensity of the sport (as categorized by METs). This is an important result, as there is a paucity of data in the literature regarding the general influence of sport intensity on HGS, as the majority of the studies have generally focused on a specific sport. Among females, there was a significantly higher amount of sport in the highest tercile of HGS. Thus, basically, a greater amount of sport does not lead to higher HGS in men, while a greater amount of sport in women similar in entity to males can lead to an increase in strength which is closer to the maximum strength, due to their relatively smaller muscle mass [104].

Considering the complexity of HGS associations with anthropometric variables, weight status, body composition, and sex, we applied multiple regression analysis in such a way that the final model retained only the most significant factors after a stepwise exclusion procedure. According to the model, body composition parameters and indices have proven to be the most effective, jointly with sex, in explaining the variability of D HGS, which was best predicted by sex, D UMA, D AFI, FM, and FFM. These factors account for about 75% of the HGS variability of the dominant side in our sample of healthy Italian young adults.

In general, it can be deduced that HGS differs significantly between sexes and is directly influenced by body composition. Consequently, we stress the impact that severe undernutrition, with subsequent deficits in FFM, can have on HGS. Based on the results obtained, although we did not have severely undernourished subjects in our sample, we can confirm that hand grip strength is an essential parameter for indicating a deteriorating nutritional status [105,106]. Norman et al. [107] suggested that a reduction in nutritional intake translates into a compensatory loss of whole-body protein, which is preferably lost from the muscle mass; that is, the largest protein reserve in the body. As a result of this process, cellular changes lead to a decrease in protein synthesis and an increase in proteolysis, thus causing fiber atrophy. This, in turn, leads to a decrease in muscle strength and muscle function. This underlines the importance of HGS, at all ages, as a marker of nutritional status which can be used to identify patients at risk of undernutrition and malnutrition and to control them after nutritional interventions. Moreover, it has been proven that the measurement of HGS could motivate malnourished subjects to improve their nutritional status and may increase their perceived quality of life [104,108].

The use of grip strength estimates is a robust method to assess muscle weakness and can be of very practical use in evaluating ageing populations, patients with neuromuscular disorders, and patients during rehabilitation or return to work or sport. Further longitudinal studies may provide relevant indications on HGS decline with body composition changes during the ageing process.

This study has some strengths and limitations. Our sample of healthy university students cannot be considered representative of the whole population, even though it has proved adequate to draw relevant conclusions. Moreover, the sample was homogeneous with respect to age, preventing us from assessing changes in HGS with the age of the participants. It must also be considered that our sample consisted only of Italians. Further investigations could be carried out in other populations, in order to gain new data and generate more reference values. The strengths of our study included the direct measurement of several anthropometric characters collected by expert operators and the quantitative assessment of handedness using recognized methods of investigation.

## 5. Conclusions

In this study, we successfully analyzed HGS according to laterality, as well as anthropometric parameters and indices, in a sample of healthy young adults, in order to obtain an appropriate model of the relationships between these variables. With regards to laterality, the dominant hand was significantly stronger in both sexes, with ambidextrous subjects showing higher HGS in the right hand. While the parameters connected with physical activity did not seem to show any association with HGS, we identified body composition parameters and sex to be the major predictors of strength in the dominant hand. Our findings are of particular relevance, suggesting that HGS can be associated with unhealthy conditions—in particular, those related to low values of FFM—and can be used to monitor changes in body composition, given that the dominant side is always measured. In conclusion, we have highlighted the importance of the application of appropriate measurement methodologies and the objective assessment of handedness, although further research is needed to provide evidence of the effectiveness and clinical relevance of HGS testing in the evaluation and prediction of critical health conditions.

## Figures and Tables

**Table 1 ijerph-17-04273-t001:** Anthropometric characteristics, sports habits, and fat and weight status by sex (R = right; L = left; D = dominant side).

Anthropometric Traits	Males		Females		
	Mean	SD	Mean	SD	*p*
Stature (cm)	178.0	7.0	163.6	6.1	<0.0001
Weight (kg)	74.9	11.1	58.2	8.0	<0.0001
BMI (kg/m^2^)	23.6	2.9	21.7	2.6	<0.0001
WC (cm)	79.7	7.8	69.3	5.4	<0.0001
L MUAC (cm)	30.4	3.5	26.5	2.8	<0.0001
R MUAC (cm)	30.7	3.3	26.6	2.7	<0.0001
D MUAC (cm)	30.6	3.3	26.6	2.8	<0.0001
L Triceps skinfold (mm)	9.8	4.8	17.0	5.1	<0.0001 ^a^
R Triceps skinfold (mm)	9.9	5.0	16.9	5.2	<0.0001 ^a^
D Triceps skinfold (mm)	9.9	4.9	16.9	5.1	<0.0001 ^a^
D TUA (cm^2^)	75.4	16.6	56.5	12.4	<0.0001
D UMA (cm^2^)	61.2	14.8	36.1	7.6	<0.0001
D UFA (cm^2^)	14.4	7.2	20.4	7.5	<0.0001
D AFI (%)	18.9	7.7	35.5	8.0	<0.0001
%F	14.4	4.4	25.8	4.5	<0.0001
FM (kg)	11.0	4.5	15.2	4.3	<0.0001
FFM (kg)	63.9	8.3	43.0	4.9	<0.0001
L HGS (kg)	43.9	8.1	27.5	5.0	<0.0001
R HGS (kg)	45.9	8.3	28.8	4.8	<0.0001
D HGS (kg)	45.7	8.2	28.9	4.7	<0.0001
D HGS/weight	0.6	0.1	0.5	0.1	<0.0001
**Sports and PA**					
Sport amount (h/week)	7.0	4.0	6.1	4.1	0.0169
Sport practice (years)	9.2	5.2	8.9	5.2	0.5458
PA (METs)	4827.6	3268.7	3621.4	3300.5	0.0005
	**N**	**%**	**N**	**%**	***p***
**Weight status**					<0.0001
Under weight	5	1.4	16	8.5	
Normal weight	269	75.6	155	82.4	
Overweight	70	19.7	15	8.0	
Obese	12	3.4	2	1.1	
**Fat status**					0.0065
Under fat	23	6.5	26	13.8	
Normal fat	295	82.8	152	80.9	
Overfat	34	9.6	10	5.3	
Very overfat	4	1.1	0	0.0	
**Distribution by categories of sports by METs**					0.9595
METs < 2	11	3.1	6	3.3	
2 ≤ METs < 4	8	2.3	3	1.6	
4 ≤ METs < 6.5	74	20.8	40	21.2	
METs ≥ 6.5	262	73.8	139	73.9	
**Distribution by categories of PA by METs**					0.0463
Light	13	3.6	11	5.6	
Moderate	70	19.8	54	28.9	
Intense	273	76.7	123	65.5	

^a^ Comparisons were performed using Log skinfolds. Abbreviations: BMI, body mass index; WC, waist circumference; MUAC, mid-upper arm circumference; TUA, total upper arm area; UMA, upper arm muscle area; UFA, upper arm fat area; AFI, arm fat index; %F, percent fat mass; FM, fat mass; FFM, fat-free mass; HGS, handgrip strength; PA, physical activity; MET, metabolic equivalent of task.

**Table 2 ijerph-17-04273-t002:** R scores (mean and SD) and frequencies of hand preference (according to Edinburgh Handedness Inventory (EHI)) by sex.

Handedness	Males		Females		
	Mean	SD	Mean	SD	*p*
R score	45.0	42.7	54.2	37.7	0.0347
**Frequencies**	**N**	**%**	**N**	**%**	0.0075
Right-handed	212	59.6	137	72.9	
Left-handed	18	5.1	8	4.3	
Ambidextrous	126	35.4	43	22.9	

**Table 3 ijerph-17-04273-t003:** Comparison between sides in handgrip strength for each handedness category by sex.

Handedness Category	Right Handgrip Strength		Left Handgrip Strength		
	Mean	SD	Mean	SD	*p*
**Males**					
Right-handed	46.0	8.8	43.6	8.4	<0.0001 ^a^
Left-handed	40.9	6.3	43.3	6.3	0.0468 ^b^
Ambidextrous	46.6	7.9	44.6	8.3	<0.0001 ^a^
**Females**					
Right-handed	29.0	4.9	27.4	5.2	<0.0001 ^a^
Left-handed	25.3	5.8	26.6	4.9	0.2489 ^b^
Ambidextrous	28.8	4.1	28.0	4.5	0.0693 ^a^

^a^ Comparison was performed using *t*-test for dependent sample; ^b^ comparison was performed using Wilcoxon non-parametric test.

**Table 4 ijerph-17-04273-t004:** Comparison between first and last tercile of dominant handgrip strength in males (D = dominant side).

Anthropometric Traits (Males)	1st Tercile (Strength ≤ 42.0 kg)		3rd Tercile (Strength ≥ 49.5 kg)		
	Mean	SD	Mean	SD	*p*
Stature (cm)	176.0	6.6	180.5	6.6	0.0000
Weight (kg)	69.5	8.8	80.9	10.9	0.0000
BMI (kg/m^2^)	22.4	2.5	24.8	2.9	0.0000
WC (cm)	76.8	6.8	82.1	8.6	0.0000
D Triceps skinfold (mm)	9.7	4.6	9.9	5.8	0.1969 ^a^
D MUAC (cm)	28.9	2.8	32.5	3.2	0.0000
D TUA (cm^2^)	67.5	13.1	85.0	17.3	0.0000
D UMA (cm^2^)	54.5	11.1	70.2	15.9	0.0000
D UFA (cm^2^)	13.3	6.1	15.1	8.6	0.0647
D AFI (%)	19.3	7.2	17.6	8.3	0.0963
%F	14.0	4.3	14.4	4.7	0.5185
FM (kg)	9.9	3.8	11.9	5.2	0.0008
FFM (kg)	59.6	6.5	69.2	7.8	0.0000
**Sports and PA**					
Sport amount (h/week)	6.6	3.7	7.4	4.3	0.1149
Sport practice (years)	10.1	5.1	8.2	5.2	0.0062
PA (METs)	4431.9	2912.9	4853.0	3050.2	0.3625
		%		%	*p*
**Weight status**					0.0001
Underweight	4	3.2	0	0.0	
Normal weight	103	83.2	75	63.6	
Overweight	16	12.8	36	30.5	
Obese	1	0.8	7	5.9	
**Fat status**					0.8168
Under fat	10	8.0	7	6.0	
Normal fat	102	82.4	95	81.2	
Overfat	11	8.8	14	12.0	
Obese	1	0.8	1	0.9	
**Distribution by categories of sports by METs**					0.0038
METs < 2	4	3.2	4	3.4	
2 ≤ METs < 4	0	0.0	4	3.4	
4 ≤ METs < 6.5	18	14.5	35	29.7	
METs ≥ 6.5	102	82.3	75	63.6	
**Distribution by categories of PA by METs**					0.7627
Light	3	2.7	2	2.1	
Moderate	28	22.7	23	19.1	
Intense	93	74.7	93	78.7	
**Handedness categories**					0.2947
Right-handed	75	60.8	76	64.4	
Left-handed	10	8.0	4	3.4	
Ambidextrous	39	31.2	38	32.2	

^a^ Comparisons were performed using Log skinfolds. Abbreviations: BMI, body mass index; WC, waist circumference; MUAC, mid-upper arm circumference; TUA, total upper arm area; UMA, upper arm muscle area; UFA, upper arm fat area; AFI, arm fat index; %F, percent fat mass; FM, fat mass; FFM, fat-free mass; PA, physical activity; MET, metabolic equivalent of task.

**Table 5 ijerph-17-04273-t005:** Comparison between the lowest and the highest tercile of dominant handgrip strength in females (D = dominant side).

Anthropometric Traits (Females)	1st Tercile (Strength ≤ 26.8 kg)		3rd Tercile (Strength ≥ 30.5 kg)		
	Mean	SD	Mean	SD	*p*
Stature (cm)	162.0	6.4	165.1	6.2	0.0062
Weight (kg)	55.6	6.1	61.8	8.7	0.0000
BMI (kg/m^2^)	21.2	2.1	22.6	2.8	0.0015
WC (cm)	68.2	4.4	70.7	5.2	0.0037
D Triceps skinfold (mm)	15.5	4.8	18.0	4.9	0.0026 ^a^
D MUAC (cm)	25.7	2.1	27.5	2.8	0.0002
D TUA (cm^2^)	52.9	8.9	60.5	13.4	0.0007
D UMA (cm^2^)	34.2	6.4	38.4	8.3	0.0036
D UFA (cm^2^)	18.7	6.0	22.1	7.8	0.0107
D AFI (%)	35.0	8.4	36.2	7.3	0.4105
%F	25.2	4.5	26.6	4.0	0.0597
FM (kg)	14.1	3.4	16.7	4.5	0.0006
FFM (kg)	41.5	4.3	45.1	5.1	0.0000
**Sports and PA**					
Sport amount (h/week)	5.4	3.6	7.0	3.7	0.0153
Sport practice (yrs)	9.0	5.6	9.1	5.2	0.9536
PA (METs)	2910.4	3080.9	3908.1	3373.6	0.1571
		%		%	*p*
**Weight status**					0.3834
Under weight	4	6.9	4	6.0	
Normal weight	51	87.9	54	80.6	
Overweight	3	5.2	7	10.4	
Obese	0	0.0	2	3.0	
**Fat status**					0.1797
Under fat	11	18.3	5	7.5	
Normal fat	45	78.3	59	88.1	
Overfat	2	3.3	3	4.5	
Obese	0	0.0	0	0.0	
**Distribution by categories of sports by METs**					0.4536
METs < 2	1	1.8	2	3.0	
2 ≤ METs < 4	2	3.5	0	0.0	
4 ≤ METs < 6.5	9	15.8	12	17.9	
METs ≥ 6.5	46	78.9	53	79.1	
**Distribution by categories of PA by METs**					0.0948
Light	3	5.9	4	6.8	
Moderate	22	38.2	14	20.3	
Intense	32	55.9	49	72.9	
**Handedness categories**					0.1770
Right-handed	39	67.2	49	73.1	
Left-handed	5	8.6	1	1.5	
Ambidextrous	14	24.1	17	25.4	

^a^ Comparisons were performed using Log skinfolds. Abbreviations: BMI, body mass index; WC, waist circumference; MUAC, mid-upper arm circumference; TUA, total upper arm area; UMA, upper arm muscle area; UFA, upper arm fat area; AFI, arm fat index; %F, percent fat mass; FM, fat mass; FFM, fat-free mass; PA, physical activity; MET, metabolic equivalent of task.

**Table 6 ijerph-17-04273-t006:** Pearson correlation coefficients between anthropometric variables and dominant handgrip strength by sex (asterisks denote statistical significance at the *p* < 0.05 level; D = dominant side).

Dominant Handgrip Strength	Males	Females
Stature	0.229 *	0.245 *
Weight	0.436 *	0.392 *
BMI	0.372 *	0.293 *
WC	0.272 *	0.270 *
D MUAC	0.502 *	0.319 *
D Triceps skinfold (Log)	−0.047	0.201 *
D TUA	0.501 *	0.317 *
D UMA	0.522 *	0.275 *
D UFA	0.089	0.232 *
D AFI	−0.125 *	0.076
%F	0.037	0.162
FM	0.176 *	0.300 *
FFM	0.490 *	0.375 *

Abbreviations: BMI, body mass index; WC, waist circumference; MUAC, mid-upper arm circumference; TUA, total upper arm area; UMA, upper arm muscle area; UFA, upper arm fat area; AFI, arm fat index; %F, percent fat mass; FM, fat mass; FFM, fat-free mass.

**Table 7 ijerph-17-04273-t007:** Results of multiple linear regression with backward regression analysis.

Predictor Variables	Dominant Handgrip Strength		
	β	t	*p*
Sex (male)	0.1980	3.4323	0.0007
D UMA	0.3463	5.9100	0.0000
D AFI	0.1345	2.1342	0.0335
FM	−0.1784	−3.6116	0.0003
FFM	0.4420	6.2158	0.0000
R^2^	0.7491		
R^2^ adjusted	0.7457		
*p*	0.0000		

β: standardized regression coefficient. Abbreviations: D UMA, upper arm muscle area (dominant side); D AFI, arm fat index (dominant side); FM, fat mass; FFM, fat-free mass.

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
