# Peer review of "Handgrip Strength in Young Adults: Association with Anthropometric Variables and Laterality"

_ijerph, 2020, doi:10.3390/ijerph17124273_

Round 1

Reviewer 1 Report

Methodology

Line 91: ...”included subjects, aged 18-30..”, However, in the line 95: “those aged > 30 were exclude”. I guess the correct description would be included from 18-29 year old, and not including (not excluding) >30 y.o or foreigners.

Question: Is there any particular reason not including the foreigners, e.g. Germans, British’s, Polishes? It definitely doesn’t sound good. To my view it should be reevaluated. I don’t see any reason that makes different a non-Italian adolescent student from Bologna’s University, taking the same classes and following the same daily routines just like any Italian student in that Campus, to not participate of the study.

Question 2: Were the people not matching the requisites excluded after being recruited or not included from the beginning?

Conclusion

The results are interesting, although variated. The correlations related to health status, the main ones BMI, WC and FM are low-moderated correlated and are not related to unhealth conditions, once the first quartile in HGS (for both sexes) despite lower HGS have better BMI, WC and FM measures, which actually points to the opposite direction.

The Prospective Urban Rural Epidemiology (PURE) study (Leong, et al., 2015) had participants with average of 50 year old, and several nationalities/incomes, and a follow-up study which made possible distinguish in many ways the correlation between grip strength and prediction of death, mainly considering there were include participants independent of health status, i.e. smokers, high blood pressure, diabetics, cancer, alcoholics, etc. Which means that, studying a health young population in a transversal study is going to show different outcomes as well as interpretation.

The purpose of the study was to correlate the grip strength status with anthropometric variables in a homogenous nationality populations, however not exactly with health/unhealth parameters, once all participants were healthy and young and lower BMIs, WC and FM were actually in the “weaker”.  I suggest changing the conclusion in order to better match the found results.   

Author Response

Methodology

R1: Line 91: ...”included subjects, aged 18-30..”, However, in the line 95: “those aged > 30 were exclude”. I guess the correct description would be included from 18-29 year old, and not including (not excluding) >30 y.o or foreigners.

Answer (A): Correct, thank you. Since the age range of our sample is 18-30 years, we have changed the symbol of the subjects to exclude from ≥ 30 to  >30  at line 101, page 3.

R1: Question: Is there any particular reason not including the foreigners, e.g. Germans, British’s, Polishes? It definitely doesn’t sound good. To my view it should be reevaluated. I don’t see any reason that makes different a non-Italian adolescent student from Bologna’s University, taking the same classes and following the same daily routines just like any Italian student in that Campus, to not participate of the study.

A: Among the foreign students, there were also non-Caucasian. Thus, and because the anthropometric traits under examination are affected by the characteristics of the population of origin, we decided to study an homogeneous sample consisting only of Italians. We have changed the text to better explain this concept (lines 96-97, page 3).

R1: Question 2: Were the people not matching the requisites excluded after being recruited or not included from the beginning?

A: people not matching the requisites were excluded after being recruited.

Conclusion

R1: The results are interesting, although variated. The correlations related to health status, the main ones BMI, WC and FM are low-moderated correlated and are not related to unhealth conditions, once the first quartile in HGS (for both sexes) despite lower HGS have better BMI, WC and FM measures, which actually points to the opposite direction.

A: Among the main characters directly related to health status, we would also consider FFM (fat free mass) and MUAC (mid-upper arm circumference), which showed higher correlation coefficient with HGS. As you observed, the 1st tercile shows, on average, no unhealthy condition for these traits. After all, we chose from the beginning to study a healthy sample of the population. However, the significantly lower values of weight, BMI, FM, FFM, MUAC, which are found in participants with lower HGS, prefigure, for instance, what is determined in case of malnutrition (undernutrition) when the reduced values of these traits are accompanied by a reduction in muscle strength. In fact, the reduction in strength is not achieved in the case of an increase in adiposity, but in the case of a decrease in adiposity. As a confirmation of this observation, comparing the weakest with the strongest individuals of our sample (first vs last tertile), there are 8.1% vs 6% underfat among males, while among females there are 19% vs 7.5% underfat.

R1: The Prospective Urban Rural Epidemiology (PURE) study (Leong, et al., 2015) had participants with average of 50 years old, and several nationalities/incomes, and a follow-up study which made possible distinguish in many ways the correlation between grip strength and prediction of death, mainly considering there were include participants independent of health status, i.e. smokers, high blood pressure, diabetics, cancer, alcoholics, etc. Which means that, studying a health young population in a transversal study is going to show different outcomes as well as interpretation.

A: thank you for this interesting comment.

R1:The purpose of the study was to correlate the grip strength status with anthropometric variables in a homogenous nationality populations, however not exactly with health/unhealth parameters, once all participants were healthy and young and lower BMIs, WC and FM were actually in the “weaker”.  I suggest changing the conclusion in order to better match the found results.   

A: as we have pointed out above, it is possible to catch a glimpse of specific anthropometric character patterns in this healthy sample as well. In any case, we have modified the conclusion as suggested (line 415-418, page 12).

Reviewer 2 Report

Abstract - several acronyms are used without definition. Please define these in the abstract or re-word so that the acronyms are not used.

Table 2 - Mean value for Females should be aligned with the other data in that row.

Table 4 - there is a floating end parenthesis the column label "Anthropometric traits (Males))"

Author Response

R2: Abstract - several acronyms are used without definition. Please define these in the abstract or re-word so that the acronyms are not used.

Answer (A): we have removed the acronyms, as suggested (line 22, page 1).

R2: Table 2 - Mean value for Females should be aligned with the other data in that row.

A: done.

R2: Table 4 - there is a floating end parenthesis the column label "Anthropometric traits (Males))"

A: done.

Reviewer 3 Report

Abstract

The authors should avoid using acronyms, or at least state what they mean (e.g. UMA, AFI, FM, etc.).

Introduction

General comments:

This section on the assessment on handgrip in young adults poor as it does not provide a clear rationale for the study. Has previous research been done on this population? And why is there a need for your study?

Additionally, please provide some hypotheses and explore the influence of handgrip in young adults on health-related outcomes.

Specific comments:

Page 1, line 31-33: this is a very tenuous statement. Needs reference.

Page 1, line 39-41: confusing sentence. Please revise.

Page 1, line 43-44: why could it be used as motivation? How can handgrip be associated with motivation (or lack of it)?

Page 2, line 45-46: correlations and effects are different statistical test. The authors should be cautious when comparing different tests and discuss them as the same.

Page 2, line 75-76: the word body size should be revised since body size is related to mass and not length, and “etc” should be avoided in scientific writing. The authors are recommended to explain in detail their rationale rather than proposing examples.

Page 2, line 76, provide a full word for BMI.

Method

Participants

Page 3, line 94: why only Italian? The authors do not provide a rationale for conducting this study only with native Italian individuals. Do they have different handgrip force compared to other nations?

Please provide some more details of the sample characteristics (e.g., students, self-employed workers, household, income).

The authors are recommended to provided an ethical committee registration number which provides more confidence to the reader.

The sample size seems to be way-off tended to males. Could this have influenced your results?

Page 3, line 99: remove “s” from females.

Data collection

Page 3, line 106: please remove “ly” from firstly

Page 3, line 123: stature? What does this word mean? Do you mean height?

Page 3-4, line 139-147: There seems to go to much in this section. The authors do not provide a rationale as to why all those parameters are needed. Please provide detailed literature as to why these parameters could influence HGS and why should we assess these indicators.

I recommend the authors to choose and analyze only those parameters that seem to be relevant for HGS.

Statistical analysis

Why the S-W test? Your data is way above the 50-cutoff point.

Is your data normally distributed to conduct parametric t-tests between gender, and tercile?

The authors also report using the K-W ANOVA across handedness groups. Why a non-parametric test? The authors do not provide any data or information concerning the normality of the data. Please explain.

And now we are back to parametric tests for correlation analysis.

It seems to me that the authors intend to do as many tests as possible with all the data available. However, there is too much going on confusing the reader on what the authors intend to exam.

Some questions that could help the authors revise their analyses:

Why all those tests?

What is the purpose of each?

Did the authors conduct a priori power analysis to work out your desired sample size? If no why not?

How do the authors intend to exam the association of sex and HGS using regression analysis?

Results

This section should be revised. First, because of the choice of the statistical test, which I have some doubts about the author's own understanding of their selection. Second, there is so much information, lacking flow and structure, as well as clarity.

I advise the authors to choose only those variables and test that seem to follow current literature on HGS assessment, rather than collecting as much data as possible and then use any test possible.

Discussion

General comments:

I will not comment much on the Discussion as I believe it will change considerably after carefully reviewing their statistical analyses.

I advise the authors on beginning this section by providing the study’s objective to the specific research question (from the introduction section) or what was intended to do. Then discuss the results based on the reported findings by the authors. Overall, it needs to be less of a summary of the results and more about the implications and interpretation of the findings.

I think the authors should consider the limitations of their research more thoroughly. As it stands, I can only see one limitation, when there are several that should be addressed (e.g., Italian specific population, sport specificity of some individuals, young adults who are students? Or workers?)

What implications does this have in the young adult world? To understand the associations between several anthropometric parameters and HGS?

Specific comments:

Page 10, line 298-300: this paragraph is just confusing. What do the authors intend to explain? Please revise

Page 11, line 347: avoid using causal language when reporting associations.

Page 11, line 353-357: did the authors collect data from the type of sport (I believe so as you calculated the METs based on the activity). Hence, why not use this information to support your statements?

Page 11, line 361: nutritional status? I do not see any variables related to nutrition. Are you referring to BMI, FM, FFM? Please revise.

Page 11, line 368-376: I am questioning, so? We all know that normal-weight individuals or those with PA have greater HGS. What are the implications in general life and/or perceived quality of life? This paragraph should be revised.

Conclusion

Please sum up what was found and how it advances current literature on the HGS assessment and the association with different anthropometric parameters. In its current form, there is insufficient explanation of the theoretical and practical consequences of the findings, or of the specific ways in which they contribute to the literature on the topic.

Specific comments.

The English writing in the current manuscript is not for publication-quality and requires major improvements. There are frequent grammatical and typographical errors present throughout the paper.

Please revise the reference list. there are several errors that need to be addressed.

Author Response

Abstract

R3: The authors should avoid using acronyms, or at least state what they mean (e.g. UMA, AFI, FM, etc.).

Answer (A): done (line 22, page 1).

Introduction

General comments:

R3: This section on the assessment on handgrip in young adults poor as it does not provide a clear rationale for the study. Has previous research been done on this population? And why is there a need for your study?

A: Following your indication, we added some sentences to let better understand the rationale of this study, as well as quotations of previous similar works regarding young adults (line 33, page 1; lines 45-47, page 2). We also shifted some sentences (lines: 59-66, page 2) to provide a clear rationale.

R3: Additionally, please provide some hypotheses and explore the influence of handgrip in young adults on health-related outcomes.

A: We are confident that we have addressed this point by introducing changes in the introduction and by better explaining the rationale of the manuscript

Specific comments:

R3: Page 1, line 31-33: this is a very tenuous statement. Needs reference.

A: We have added appropriate references (line 33, page1).

R3: Page 1, line 39-41: confusing sentence. Please revise.

A: The sentence has been rewritten (lines 39-41, page 1).

R3: Page 1, line 43-44: why could it be used as motivation? How can handgrip be associated with motivation (or lack of it)?

A: We added indications on the association between handgrip and motivation (line 42-44,page 1).

R3: Page 2, line 45-46: correlations and effects are different statistical test. The authors should be cautious when comparing different tests and discuss them as the same.

A: The term “correlation” has been changed with “relationship”(line 48,page 2).

R3: Page 2, line 75-76: the word body size should be revised since body size is related to mass and not length, and “etc” should be avoided in scientific writing. The authors are recommended to explain in detail their rationale rather than proposing examples.

A: The term body size has been changed with body dimensions (line 60,page 2), and the rationale of the study has been explained in detail (as reported above).

R3: Page 2, line 76, provide a full word for BMI.

A: The full word for BMI has been provided (line 60,page 2).

Method

Participants

R3: Page 3, line 94: why only Italian? The authors do not provide a rationale for conducting this study only with native Italian individuals. Do they have different handgrip force compared to other nations?

A:  Since anthropometric traits are affected by the characteristics of the population of origin (as now highlighted on lines 96-97, page 3), we have chosen the largest and most homogenous subsample in our sample.

R3: Please provide some more details of the sample characteristics (e.g., students, self-employed workers, household, income).

A: all the participants were recruited among the students of the University of Ferrara and Bologna, as reported at lines 107-108, page 3.

R3: The authors are recommended to provided an ethical committee registration number which provides more confidence to the reader.

A: done (lines 116-117, page 3).

R3: The sample size seems to be way-off tended to males. Could this have influenced your results?

A: No, because we have done all the computations on sex-disaggregated data.

R3: Page 3, line 99: remove “s” from females.

A: In the indicated line, the word “females” is repeated twice, but the plural is required in both.  

Data collection

R3: Page 3, line 106: please remove “ly” from firstly

A: done. 

R3: Page 3, line 123: stature? What does this word mean? Do you mean height?

A: Stature is the distance from the ground to the highest point of the head (vertex) in an upright position, with the landmark taken along the median sagittal plane of the head oriented in the Frankfurt plane. The word “height”, although widely used, is a generic term in Anthropometry. There are many heights, such as shoulder height (akromion height), eye height (ectocanthion height),  etc., but only one stature. Therefore, if the landmark specification is not added (vertex height), the term stature is more appropriate.

R3: Page 3-4, line 139-147: There seems to go to much in this section. The authors do not provide a rationale as to why all those parameters are needed. Please provide detailed literature as to why these parameters could influence HGS and why should we assess these indicators.

A: all these anthropometric parameters are commonly used in literature to describe the characteristics of the body with particular regard to individual weight status and adiposity (among many others: WHO, 1995; Frisancho, 2008; Kulathinal et al, 2016; Madden and Smith, 2016; Gualdi-Russo et al, 2016; Tur and Bibiloni, 2019). The reasons for the analysis of these anthropometric characters in relation to HGS were given in the introduction, with details on the reference literature. The analysis of the association of HGS with anthropometric variables is a main goal of our study, as highlighted in the title of the manuscript. Why all these parameters? Because these different anthropometric parameters analyse distinct aspects of the human body, which are complementary to each other and allow a greater understanding of what a single parameter (e.g., the abused BMI) can provide.

R3: I recommend the authors to choose and analyze only those parameters that seem to be relevant for HGS.

A: We are confident that all the selected parameters examined are of relevance in this study, as reported above, according to literature and to our well-founded experience in the field.

Statistical analysis

R3: Why the S-W test? Your data is way above the 50-cutoff point.

A: Thank you for this indication. Indeed, given the size of the total sample, we agree with you that this test is not necessary. We have therefore deleted the sentence (line 165, page 4).

R3: Is your data normally distributed to conduct parametric t-tests between gender, and tercile?

A: also in this case, our data are way above the 50-cutoff point. 

R3: The authors also report using the K-W ANOVA across handedness groups. Why a non-parametric test? The authors do not provide any data or information concerning the normality of the data. Please explain.

A: given the size of the sample, we repeated the analysis using the parametric statistics (line 174, page 4)  

R3: And now we are back to parametric tests for correlation analysis.

A: There was no violation of the assumption of parametric statistics in this case. Following your suggestions, we have now used non-parametric tests only when we compared small sub-samples (e.g., left handed).

R3: It seems to me that the authors intend to do as many tests as possible with all the data available. However, there is too much going on confusing the reader on what the authors intend to exam.

A: sorry if we gave you this impression, we can assure that we had only intended to apply the more appropriate statistics. Therefore, we gladly corrected the statistics where it was appropriate to do so, according to your suggestions.

R3: Some questions that could help the authors revise their analyses:

Why all those tests?

 What is the purpose of each?

 Did the authors conduct a priori power analysis to work out your desired sample size? If no why not?

 How do the authors intend to exam the association of sex and HGS using regression analysis?

 A: Thank you very much for trying to help us improve the work with these additional questions. We asked ourselves the same questions before comparing subgroups (sex, handedness category, etc.). We have carefully chosen the statistical tests according to our objectives and taking into account the size of the sub-samples. With regard to sampling, we took a convenience sample that included all the students who volunteered to participate in the study at the universities of Bologna and Ferrara, according to our inclusion and exclusion criteria.

The categorical variable “sex” with two levels (0 for males; 1 for females) entered as predictor in the multiple regression analysis in addition to the quantitative variables in order to examine the association with HGS.

Results

R3: This section should be revised. First, because of the choice of the statistical test, which I have some doubts about the author's own understanding of their selection. Second, there is so much information, lacking flow and structure, as well as clarity.

A: We agree with you on the choice of parametric tests instead of non-parametric ones, so we replaced the non-parametric K-W Anova and M-W Test with the corresponding parametric tests (see Table 2, page 6; lines 218-219, page 6).

The main goals of our study were to assess the association of laterality and anthropometric traits with hand muscle strength in young adults. In the manuscript, we started with a descriptive analysis, then we made a comparison between sexes, handedness, terciles of HGS, correlation between HGS and anthropometric traits, and finally we applied the multiple regression analysis to highlight the determinants of the dominant hand strength in our sample of young adults. In our opinion, all these steps are necessary to reach our goals.

R3:I advise the authors to choose only those variables and test that seem to follow current literature on HGS assessment, rather than collecting as much data as possible and then use any test possible.

A: For this study, we selected only the anthropometric traits that possibly could be related to HGS on the basis of current literature and our opinion as experienced anthropometrists (as explained above).

Discussion

General comments:

R3:I will not comment much on the Discussion as I believe it will change considerably after carefully reviewing their statistical analyses.

A: We have careful checked the statistical analysis performed according to your comments. We think that now the tests used are more appropriate and all of them are useful to achieve the objectives of the study. Nevertheless, we could confirm our previous results and the discussion in a whole.

R3: I advise the authors on beginning this section by providing the study’s objective to the specific research question (from the introduction section) or what was intended to do. Then discuss the results based on the reported findings by the authors. Overall, it needs to be less of a summary of the results and more about the implications and interpretation of the findings.

A: Thank you for your suggestion. We have now added the aims of the study at the beginning of the discussion section (line 272-276, page 9).

R3: I think the authors should consider the limitations of their research more thoroughly. As it stands, I can only see one limitation, when there are several that should be addressed (e.g., Italian specific population, sport specificity of some individuals, young adults who are students? Or workers?)

A: According to your suggestions, we revised this section (lines 397, 400-402, page 12). As regards sports, we agree with you on the importance of the specificity of each sport on HGS as we had discussed at lines 363-367. However, and according to international recommendations, in order to avoid excessive sample fragmentation, we have categorized sports on the basis of their intensity, expressed as METs.

R3: What implications does this have in the young adult world? To understand the associations between several anthropometric parameters and HGS?

A: we discussed general implications at lines 387-392 page 11, as suggested.

Specific comments:

R3: Page 10, line 298-300: this paragraph is just confusing. What do the authors intend to explain? Please revise

A: we have now revised the paragraph (line 307-313, page 10)

R3: Page 11, line 347: avoid using causal language when reporting associations.

A: we have rewritten the sentence avoiding causal language (line 359-360, page 11)

R3:Page 11, line 353-357: did the authors collect data from the type of sport (I believe so as you calculated the METs based on the activity). Hence, why not use this information to support your statements?

A: You are right, we collected the data regarding the type of sport practiced from each subjects. However, in this study, we decided to analyse the association between sports intensity and HGS rather than sports specificity for the reasons stated above.

R3: Page 11, line 361: nutritional status? I do not see any variables related to nutrition. Are you referring to BMI, FM, FFM? Please revise.

A: We have revised the sentence (line 372-373, page 11)

R3: Page 11, line 368-376: I am questioning, so? We all know that normal-weight individuals or those with PA have greater HGS. What are the implications in general life and/or perceived quality of life? This paragraph should be revised.

A: we have now add the possible implications in lines 383, page 11

Conclusion

R3: Please sum up what was found and how it advances current literature on the HGS assessment and the association with different anthropometric parameters. In its current form, there is insufficient explanation of the theoretical and practical consequences of the findings, or of the specific ways in which they contribute to the literature on the topic.

A: We have now modified the conclusion according to your suggestion (lines 408-411, 413-418, page 12). 

Specific comments.

R3: The English writing in the current manuscript is not for publication-quality and requires major improvements. There are frequent grammatical and typographical errors present throughout the paper.

A: the paper was revised to improve the English writing.

R3: Please revise the reference list. there are we several errors that need to be addressed.

A: we revised carefully all the reference list according to the Journal’s rules.

References:

Frisancho, A.R. Anthropometric Standards: An Interactive Nutritional Reference of Body Size and Body Composition for Children and Adults, 2nd ed.; University of Michigan Press: Ann Arbor, 2008.

Gualdi-Russo E, Rinaldo N, Khyatti M, Lakhoua C, Toselli S. Weight status, fatness and body image perception of North African immigrant women in Italy. Public Health Nutr. 2016;19(15):2743‐2751. doi:10.1017/S1368980016000872.

Kulathinal S, Freese R, Korkalo L, Ismael C, Mutanen M. Mid-upper arm circumference is associated with biochemically determined nutritional status indicators among adolescent girls in Central Mozambique. Nutr Res. 2016;36(8):835‐844. doi:10.1016/j.nutres.2016.04.007

Madden AM, Smith S. Body composition and morphological assessment of nutritional status in adults: a review of anthropometric variables. J Hum Nutr Diet. 2016;29(1):7‐25. doi:10.1111/jhn.12278.

Tur JA, Bibiloni MDM. Anthropometry, Body Composition and Resting Energy Expenditure in Human. Nutrients. 2019;11(8):1891.

World Health Organization. Physical Status: The Use and Interpretation of Anthropometry, Report of a WHO Expert Committee; World Health Organization: Geneva, Switzerland, 1995

Round 2

Reviewer 3 Report

The authors improved the manuscript. However, there are still several points that should be addressed.

Introduction

Specific comments:

Page 2, line 60: state de controversial results, so readers can understand why you considered BMI for your analyses.

Method and Results

The authors continue to not provide any data or information concerning the normality of the data. Please report values of skewness and kurtosis to help readers understand the choice of the statistical tests.

I believe I understand the use of so many indicators of HGS for your analyses. However, you should focus on what it is important to report and explain your intentions more efficiently. By this I suggest the authors to limit their analyses on those that contribute to the overall goal, rather than conducting several tests as the authors tend to report “comparison between sexes, handedness, terciles of HGS, correlation between HGS and anthropometric traits, and finally we applied the multiple regression analysis to highlight the determinants of the dominant hand strength in our sample of young adults”.

We don’t need to conduct mean and variance comparisons to conduct multiple regression analyses, nor can we interpret these results in a similar manner. Related to this, why multiple regression using D AFI when you already know that it is only correlated (small and negatively, but still significant) in the male sample? It is expected that it would not have a predictive power in the model. There are so many variables that presented statistical significance with DHS but the authors did not consider them.

The authors also propose: “predictors of HGS values ​​between different anthropometric characteristics and establish the impact of physical and sports activities on hand strength”, however, none of this is analyzed nor discussed.

Discussion

The discussion still seems a mere description of the results at little emphasis is taken on discussing what really matters and what professionals and scholars can use and apply.

As it stands, it is nothing more than a descriptive and to some extend correlational study but it doesn’t really address something terribly substantive so ultimately just provides more evidence that HGS is associated with the dominant hand, body dimensions, sex, and other variables in some sense or another established in the literature.

Conclusion

Page 12, Line 412: this is not an experimental study, thus causal language should be avoided. Neither is “nutritional intervention” a conclusion of your study, since this variable, as you could call it, was not measured nor examined.

Present the reference values you obtained from your analyses here since it was one of your main objectives.

Specific comments.

The authors report “the article was revised to improve English writing.” But I could not see any improvements. In fact, English writing in the revised manuscript did not change. There are major grammatical errors that would not emerge in this revised manuscript if it would have gone through a proof-reading editor or a native English speaker. Please consider in the next revision of your manuscript.

Author Response

Introduction

Specific comments:

R3: Page 2, line 60: state de controversial results, so readers can understand why you considered BMI for your analyses.

 A: we specified the controversial results on BMI (lines 63-66)

Method and Results

R3: The authors continue to not provide any data or information concerning the normality of the data. Please report values of skewness and kurtosis to help readers understand the choice of the statistical tests.

 A: we provided information concerning the normality of the data (lines 166-168), as suggested.

R3: I believe I understand the use of so many indicators of HGS for your analyses. However, you should focus on what it is important to report and explain your intentions more efficiently. By this I suggest the authors to limit their analyses on those that contribute to the overall goal, rather than conducting several tests as the authors tend to report “comparison between sexes, handedness, terciles of HGS, correlation between HGS and anthropometric traits, and finally we applied the multiple regression analysis to highlight the determinants of the dominant hand strength in our sample of young adults”.

A: We stress that the statistical tests here applied are required to achieve distinct purposes in our study. Consistently with the objectives of our study, after describing the anthropometric and handedness characteristics of our sample by sex, we compared the HGS of the two sides within each handedness category by sex (t-test and Wilcoxon test) and the HGS of the dominant side among the three categories of handedness (ANOVA). Then, we examined the characteristics of the tercile with greater HGS compared to the one with lower HGS by sex. Finally, after examining the associations of anthropometric variables with HGS, we performed a backward multiple regression analysis that allowed us to determine the main predictors of dominant HGS and their contribution to the total variance.

R3: We don’t need to conduct mean and variance comparisons to conduct multiple regression analyses, nor can we interpret these results in a similar manner. Related to this, why multiple regression using D AFI when you already know that it is only correlated (small and negatively, but still significant) in the male sample? It is expected that it would not have a predictive power in the model. There are so many variables that presented statistical significance with DHS but the authors did not consider them.

A: We agree that mean and variance comparisons are not necessary to carry out multiple regression analyses (we addressed this issue in our answer above).

Related to multiple regression using D AFI, there may have been a misunderstanding. We used backward multiple regression analysis, an approach that starts with all the variables, and, at each step, gradually eliminates variables from the regression model. The outcome of this statistical process is a reduced model that provides the best explanation of the data. In our case, all the anthropometric variables entered at the same time into the model, then, we obtained the reduced set of variables (which included D AFI) which is reported in Table 7. This method allowed us to select with an appropriate strategy those variables that are necessary to explain our data, helping us also to determine the level of importance of each predictor variable (in term of contribution to the total variance) and to assess its effects once the other predictor variables are statistically eliminated.

R3:The authors also propose: “predictors of HGS values ​​between different anthropometric characteristics and establish the impact of physical and sports activities on hand strength”, however, none of this is analyzed nor discussed.

A: We used METs (Metabolic Equivalent of Task) categories. METs consider the energy expenditure during physical activity and allow to classify different sports in categories. This is very useful to be applied to our dataset, because the variety of sports practiced by the participants was very wide. In addition to anthropometric variables, also METs were included in the backward regression analysis (see line 183) although they did not result to predict significantly HGS. This aspect was also addressed in the conclusions (line 440). More in general, a discussion of the results obtained in our study considering HGS in relation to physical activity and sport practice variables has been reported at lines 390-400

Discussion

R3:The discussion still seems a mere description of the results at little emphasis is taken on discussing what really matters and what professionals and scholars can use and apply.

A:  We discussed, not only described, our results comparing them with others from the scientific literature. However, we have added further details following the suggestions of the Editor.

R3:As it stands, it is nothing more than a descriptive and to some extend correlational study but it doesn’t really address something terribly substantive so ultimately just provides more evidence that HGS is associated with the dominant hand, body dimensions, sex, and other variables in some sense or another established in the literature.

A: Besides increasing the scientific knowledge on this topic, our results may provide valuable practical information for clinicians.

Conclusion

R3: Page 12, Line 412: this is not an experimental study, thus causal language should be avoided. Neither is “nutritional intervention” a conclusion of your study, since this variable, as you could call it, was not measured nor examined.

 A: We have rewritten the sentence avoiding causal language. With regard to the “nutritional intervention”, we intended to suggest a possible further implication for our study, however we have changed the sentence.

R3: Present the reference values you obtained from your analyses here since it was one of your main objectives.

A: We added the reference values in the discussion section (lines 317-318)

Specific comments.

R3: The authors report “the article was revised to improve English writing.” But I could not see any improvements. In fact, English writing in the revised manuscript did not change. There are major grammatical errors that would not emerge in this revised manuscript if it would have gone through a proof-reading editor or a native English speaker. Please consider in the next revision of your manuscript.

A: a professional language editing service reviewed the manuscript.